# Outcomes of Kidney Transplant Recipients with Sickle Cell Disease: An Analysis of the 2000–2019 UNOS/OPTN Database

**DOI:** 10.3390/jcm10143063

**Published:** 2021-07-11

**Authors:** Napat Leeaphorn, Charat Thongprayoon, Pradeep Vaitla, Panupong Hansrivijit, Caroline C. Jadlowiec, Shennen A. Mao, Api Chewcharat, Sreelatha Katari, Pattharawin Pattharanitima, Boonphiphop Boonpheng, Wisit Kaewput, Michael A. Mao, Matthew Cooper, Wisit Cheungpasitporn

**Affiliations:** 1Renal Transplant Program, University of Missouri-Kansas City School of Medicine, Saint Luke’s Health System, Kansas City, MO 64111, USA; napat.leeaphorn@gmail.com (N.L.); skatari@saint-lukes.org (S.K.); 2Division of Nephrology and Hypertension, Department of Medicine, Mayo Clinic, Rochester, MN 59005, USA; 3Division of Nephrology, University of Mississippi Medical Center, Jackson, MS 39216, USA; pvaitla@umc.edu; 4Department of Internal Medicine, University of Pittsburgh Medical Center Pinnacle, Harrisburg, PA 17101, USA; hansrivijitp@upmc.edu; 5Division of Transplant Surgery, Mayo Clinic, Phoenix, AZ 85054, USA; Jadlowiec.Caroline@mayo.edu; 6Division of Transplant Surgery, Mayo Clinic, Jacksonville, FL 32224, USA; Mao.Shennen@mayo.edu; 7Department of Medicine, Mount Auburn Hospital, Harvard Medical School, Cambridge, MA 02138, USA; api.che@hotmail.com; 8Department of Internal Medicine, Faculty of Medicine, Thammasat University, Pathum Thani 12120, Thailand; 9Department of Medicine, Icahn School of Medicine at Mount Sinai, New York, NY 10029, USA; 10Department of Medicine, David Geffen School of Medicine, University of California, Los Angeles, CA 90095, USA; boonpipop.b@gmail.com; 11Department of Military and Community Medicine, Phramongkutklao College of Medicine, Bangkok 10400, Thailand; wisit_nephro@hotmail.com; 12Division of Nephrology and Hypertension, Mayo Clinic, Jacksonville, FL 32224, USA; mao.michael@mayo.edu; 13Medstar Georgetown Transplant Institute, Georgetown University School of Medicine, Washington, DC 20007, USA; Matthew.Cooper@gunet.georgetown.edu

**Keywords:** sickle cell disease, sickle cell, kidney transplantation, transplantation, outcomes, big data

## Abstract

Background: Lower patient survival has been observed in sickle cell disease (SCD) patients who go on to receive a kidney transplant. This study aimed to assess the post-transplant outcomes of SCD kidney transplant recipients in the contemporary era. Methods: We used the OPTN/UNOS database to identify first-time kidney transplant recipients from 2010 through 2019. We compared patient and allograft survival between recipients with SCD (*n* = 105) vs. all other diagnoses (non-SCD, *n* = 146,325) as the reported cause of end-stage kidney disease. We examined whether post-transplant outcomes improved among SCD in the recent era (2010–2019), compared to the early era (2000–2009). Results: After adjusting for differences in baseline characteristics, SCD was significantly associated with lower patient survival (HR 2.87; 95% CI 1.75–4.68) and death-censored graft survival (HR 1.98; 95% CI 1.30–3.01), compared to non-SCD recipients. The lower patient survival and death-censored graft survival in SCD recipients were consistently observed in comparison to outcomes of recipients with diabetes, glomerular disease, and hypertension as the cause of end-stage kidney disease. There was no significant difference in death censored graft survival (HR 0.99; 95% CI 0.51–1.73, *p* = 0.98) and patient survival (HR 0.93; 95% CI 0.50–1.74, *p* = 0.82) of SCD recipients in the recent versus early era. Conclusions: Patient and allograft survival in SCD kidney recipients were worse than recipients with other diagnoses. Overall SCD patient and allograft outcomes in the recent era did not improve from the early era. The findings of our study should not discourage kidney transplantation for ESKD patients with SCD due to a known survival benefit of transplantation compared with remaining on dialysis. Urgent future studies are needed to identify strategies to improve patient and allograft survival in SCD kidney recipients. In addition, it may be reasonable to assign risk adjustment for SCD patients.

## 1. Introduction

Sickle cell disease (SCD) is the most common inherited blood disorder in the United States, affecting approximately 100,000 Americans [1,2,3,4]. SCD occurs in one out of every 365 African-American births and 1 out of every 16,300 Hispanic-American births [1,2,3]. Patients with SCD are at risk for progressive kidney disease, initially manifesting as microalbuminuria and urine-concentrating defects in childhood with subsequent progression to overt proteinuria and progressive decline in kidney function after age 20 [1,2,3,4]. Almost half of SCD patients with HbSS genotype develop chronic kidney disease; the prevalence of end stage kidney disease (ESKD) ranges from 5% up to 18% with a median age of 23 years [1,5,6,7]. Despite the prevalence of ESKD in SCD, few of these individuals receive a kidney transplant [8,9,10].

Ongoing unequal access to kidney transplantation for SCD is a reflection of the transplant community’s concerns related to increased mortality risk for SCD patients compared to ESKD patients from other causes [11,12]. Nonetheless, despite having an increased risk, SCD ESKD patients receive a survival benefit following kidney transplantation similar to non-SCD ESKD patients [8,9]. Despite improvement in survival after transplantation in SCD patients [8,9], there are still concerns regarding worse outcomes among kidney transplant recipients with SCD compared to non-SCD recipients [8,10]. Data from prior to 2000 [8,10] including data from United States Renal Data System (USRDS) and the United Network for Organ Sharing (UNOS) databases demonstrated that kidney transplant recipients with SCD had reduced allograft and patient survival compared to non-SCD kidney transplant recipients [8,10]. Despite these concerns, a previous study using UNOS database demonstrated improved patient survival among SCD recipients over time (from years 1988–1999 to 2000–2011) [13].

Immunosuppression and kidney transplant care have significantly advanced in the past decades [14,15,16,17], and overall kidney transplantation outcomes have significant improved in the recent era [18,19]. In addition to emerging immunosuppressing agents, advances in human leukocyte antigen (HLA)/epitope matching, infection prevention, pre-transplant apolipoprotein L1 gene (APOL1), screening standardization of pre-transplant preparation for SCD patients, use of post-transplant SCD disease modifying therapy may impact transplant outcomes of patients with SCD [20,21,22]. The outcomes of kidney transplant for SCD ESKD recipients in the current transplant era remain underreported. We conducted this study using the UNOS database with the aim to (1) compare patient and allograft survival between recipients with SCD vs. non-SCD in the recent era (2010–2019), and to (2) examine outcomes of SCD recipients in the recent era (2010–2019), compared to the early era (2000–2009).

## 2. Methods

### 2.1. Data Source and Study Population

This study used the Organ Procurement and Transplantation Network (OPTN) /UNOS database. The OPTN/UNOS contains patient-level data of all United States transplant events. Institutional review board approval was waived due to publicly available nature of the de-identified OPTN/UNOS database.

The primary cohort includes pediatric and adult solitary first kidney transplant recipients occurring from 2010 to 2019. Patient and allograft survival between recipients with SCD ESKD were compared to recipients with all other diagnoses (non-SCD ESKD). In addition, we compared post-transplant outcomes of SCD recipients with the three most common causes of end-stage kidney disease: diabetes mellitus (DM), glomerular disease, and hypertension.

In the secondary cohort, pediatric and adult SCD ESKD patients receiving solitary first kidney transplants from 2000 to 2019 were identified. Post-transplant outcomes were compared between SCD recipients in two eras: an earlier era (Era 1: 2000–2009) versus recent (Era 2: 2010–2019)

### 2.2. Data Collection

The following variables were extracted: recipient age, sex, race, body mass index (BMI), transplant type, dialysis duration, causes of end-stage kidney disease, SCD status, history of diabetes mellitus, panel reactive antibody (PRA) level, donor age, kidney donor profile index (KDPI), induction and maintenance therapy. SCD as the cause of ESRD was reported by individual transplant center to OPTN/UNOS database. Missing data were not imputed.

### 2.3. Outcome Measures

The primary outcome was death-censored graft failure, defined as the need for dialysis or kidney re-transplant, with patients censored for death or at last follow-up date reported to the OPTN/UNOS database. The secondary outcome was all-cause mortality. Patients were followed until outcomes occurred, the end of study period (27 September 2019), or 5 years after kidney transplant, whichever was earlier.

### 2.4. Statistical Analysis

Baseline characteristics were summarized using medians with interquartile ranges for continuous variables or counts with percentage for categorical variables. Between-group comparison was performed using Wilcoxon rank-sum test for continuous variables and Chi-squared test for categorial variables. Kaplan–Meier plots were utilized to generate death-censored graft survival and patient survival curve. The log-rank test offered statistical comparison. Cox proportional hazard analysis was utilized to calculate hazard ratio (HR) and 95% confidence interval (95% CI) of death-censored graft failure and mortality. This association was adjusted for recipient, donor, and transplant-related factors that significantly differed between groups with *p* < 0.05. All *p*-values were two-tailed; *p*-values of < 0.05 were considered statistically significant. STATA version 14.1 (StataCorp, College Station, TX, USA) was utilized for all statistical analyses.

## 3. Results

### 3.1. Clinical Characteristics between SCD vs. Non-SCD Recipients in Recent Era

The primary cohort was comprised of a total of 146,430 patients who received a first kidney transplant from 2010 to 2019. There were 105 SCD and 146,325 non-SCD kidney transplant recipients. Of the non-SCD recipients, 40,362 (27.6%) had DM, 33,979 (23.2%) had glomerular disease, and 33,617 (23.0%) had hypertension as the reported cause of ESKD.

Table 1 compares baseline characteristics between SCD and non-SCD recipients. SCD recipients were younger, more likely to be female, African American, have a lower BMI, and less DM. They were more likely to have a higher PRA, be on dialysis at the time of transplant, and receive thymoglobulin as induction therapy. In addition, kidney donors for SCD recipients were younger, more African American, and more likely to have lower KDPI.

### 3.2. Post-Transplant Outcomes between SCD vs. Non-SCD Recipients in Recent Era

One-year death-censored graft survival in SCD was lower than in non-SCD recipients (94% vs. 98%; *p* = 0.02), whereas there was no significant difference in patient survival between SCD and non-SCD recipients (96% vs. 97%; *p* = 0.36). In adjusted analysis, SCD recipients were significantly associated with lower 1-year death-censored graft survival (HR 2.22; 95% CI 1.01–4.95). SCD recipients had lower 1-year patient survival than non-SCD recipients with HR of 2.24 although it did not reach statistical significance (*p* = 0.11) due to small number of deaths within 1 years after kidney transplant.

Five-year death-censored graft survival in SCD was lower than in non-SCD recipients (71% vs. 89%; *p* < 0.001) (Figure 1), whereas five-year patient survival was comparable between the two groups (83% vs. 87%; *p* = 0.12) (Figure 2). In adjusted analysis, SCD recipients were significantly associated with lower death-censored graft survival (HR 1.98; 95% CI 1.30–3.01, *p* = 0.001) (Table 2 A) and patient survival (HR 2.87; 95% CI 1.75–4.68, *p* < 0.001), compared to non-SCD recipients (Table 2 B).

Post-transplant outcomes were compared between SCD and other more prevalent causes of end-stage kidney disease. Five-year death-censored graft survival was 89% for DM, 89% for glomerular disease, and 88% for hypertension (Figure 3). Five-year patient survival was 78% for DM, 93% for glomerular disease, and 86% for hypertension (Figure 4). In adjusted analysis, SCD was significantly associated with lower death-censored graft survival compared to DM (HR 2.32; 95% CI 1.52–3.56, *p* < 0.001), glomerular disease (HR 1.91; 95% CI 1.25–2.92), and hypertension (HR 1.91; 95% CI 1.25–2.92, *p* = 0.003) (Table 2A). In addition, SCD was significantly associated with lower patient survival compared to DM (HR 1.96; 95% CI 1.20–3.22, *p* = 0.007), glomerular disease (HR 3.41; 95% CI 2.07–5.61, *p* < 0.001), and hypertension (HR 3.08; 95% CI 1.88–5.04, *p* < 0.001) (Table 2B).

### 3.3. Clinical Characteristics and Post-Transplant Outcomes between SCD Recipients in an Early vs. Recent Era

In the secondary cohort, a total of 233 SCD ESKD kidney transplants were performed from 2000 to 2019. There were 128 SCD ESKD kidney recipients in 2000–2009 (early era) and 105 in 2010–2019 (recent era). Table 3 compares baseline characteristics of SCD recipients between the early and recent era.

Death-censored graft survival (71% vs. 66%; *p* = 0.68) (Figure 5) and patient survival (83% vs. 78%; *p* = 0.69) (Figure 6) were comparable between SCD recipients in an early and recent era. There was no significant difference in death censored graft survival (HR 0.99; 95% CI 0.51–1.73, *p* = 0.98) and patient survival (HR 0.93; 95% CI 0.50–1.74, *p* = 0.82) between the two eras. 

## 4. Discussion

In this study utilizing the UNOS/OPTN database from 2000 to 2019, kidney transplant recipients with SCD had lower death censored graft survival and increased mortality compared to overall non-SCD recipients. There was no significant difference in the allograft and patient survival between the recent era (2010–2019) versus the early era (2000–2009).

The increased risk of allograft loss in SCD kidney transplant recipients is likely multifactorial. SCD is characterized by vaso-occlusion, especially in the kidneys [23], resulting in kidney infarction, papillary necrosis, hematuria, focal segmental glomerulosclerosis, and diabetes insipidus [23,24]. Successful kidney transplantation significantly improves kidney function among ESKD patients with SCD; however, unfortunately, the kidney allograft can still be affected by SCD [25], evidenced by iron/heme deposition in proximal renal tubules and related acute tubular injury in the kidney allograft biopsies [26,27]. In addition, chronic allograft injury and anemia may accelerate kidney function decline by limiting oxygen delivery [28]. Hypoxia may also lead to the formation of a reactive oxygen species that potentiates tissue inflammation [28]. Furthermore, allograft thrombosis during sickle cell crisis after kidney transplantation has been reported [27,29,30]. Despite improvement of overall transplant care and immunosuppression, our evaluation of the recent era (2010–2019) reveals persistent lower allograft survival among kidney transplant recipients with SCD compared to those with non-SCD, which is consistent with previously published studies before 2000 from the UNOS and USRDS databases [8,10]. Hydroxyurea and blood transfusion have been the primary methods used to treat SCD complications [20,21]. Although data on the effects of hydroxyurea on kidney transplant allograft are limited, the use of hydroxyurea has shown potential benefits in reduction of proteinuria, stabilization in kidney function, and reduced mortality risk among non-kidney transplant SCD patients [21,31,32,33]. While early post-transplant blood transfusion may increase immunological risk and increased risk of rejection, especially in under-immunosuppressed recipients [34], a recent study demonstrated that blood transfusion while on adequate immunosuppressive regimens within 1 week post-kidney transplantation is safe without significant association with de novo HLA-DSA development [35]. Among kidney transplant recipients with SCD, a well-conducted multicenter study on 34 SCD kidney transplant recipients between 1997 and 2017 across six London hospitals ensured the safety of long-term automated exchange blood transfusions (EBT) with improvement in allograft and patient outcomes and no increase in antibody formation or allograft rejection [20]. However, data on the use of hydroxyurea and blood transfusion are limited in the OPTN/UNOS database. Thus, future studies are needed to assess the impact of hydroxyurea, EBT, and perioperative management on outcomes of kidney transplant recipients with SCD. Furthermore, there are now new medications recently made available that are expected to be of significant benefit among SCD patients [36,37,38,39], which require future studies in kidney transplant recipients.

In our study of immunosuppression in the recent era, we also found a significantly reduced patient survival among kidney transplant recipients with SCD compared to non-SCD recipients. This is similar to the higher mortality found in SCD kidney recipients from published older era studies [8,10,13]. While a previous study utilizing the UNOS database demonstrated improved patient survival among SCD recipients from 2000–2011 compared with an earlier cohort of SCD recipients (1988–1999) [13], we did not find a significant improvement in patient survival and death-censored graft survival of SCD recipients in years 2010–2019 when compared to years 2000–2009. Furthermore, compared to other causes of ESKD, including diabetes, glomerular disease, and hypertension, our adjusted analysis showed that SCD was associated with the lower patient survival and death-censored graft survival. While patient survival in the prior era (2000–2011) was comparable among SCD recipients and diabetic recipients [13], our study shows better patient survival among diabetic recipients compared with SCD recipients. Given that there is no difference between patient survival among SCD in years 2010–2019 when compared to years 2000–2009, this finding likely reflects a significant improvement in diabetes care post kidney transplantation for the past decade [40,41,42]. Kidney transplant recipients with SCD still have complications of SCD post-transplant [20,21,22]. Since SCD is a systemic disease, it can result in extra-renal manifestations, such as pulmonary hypertension, thrombotic events, infections, cardiomyopathy and cirrhosis [43,44]. Unfortunately, the findings from our study suggest a lack in similar improvement in SCD care post-transplantation when compared to other causes of ESKD including diabetes. Future studies are required to identify effective strategies to improve care for kidney transplant recipients with SCD such as care in a multi-center system and long-term follow-up in tertiary care centers.

Kidney transplantation improves survival in patients with SCD after kidney transplant when compared to SCD patients on maintenance dialysis [9]. However, SCD patients have lower rates of waitlisting and transplantation rate after listing despite survival benefit with transplantation [9]. Thus, our study should not discourage kidney transplantation for ESKD patients with SCD. Rather, our findings highlight the urgent need for strategies to improve transplant outcomes among SCD kidney transplant recipients. Multidisciplinary subspecialty coordination likely offers the most benefit for management of kidney transplant recipients with SCD. Furthermore, SCD patients are less likely to be waitlisted or transplanted when compared to patients of similar age with other causes of ESKD [9]. This is potentially due to increased risk for graft loss and mortality after transplantation [8,10], or racial disparities in kidney transplantation [45,46], representing a potential “blind spot” regarding racial disparities in access and health outcomes [47]. Transplant centers are monitored for outcomes and may be sanctioned if transplant outcomes do not meet expected outcomes over a period of time. Expected outcomes are calculated based on patient population and comorbidities, which include primary diagnosis at transplant. Risk is adjusted for recipients with diabetes mellitus, hypertension and glomerular disease [48]; however, SCD is currently not considered for risk adjustment. Based on our data and previous reports [9,10], SCD recipients experience increased an risk of graft failure and mortality compared to the control group. SCD patients benefit from kidney transplant, despite increased morbidity and mortality, thus, it is reasonable to assign risk adjustment for SCD patients. This will reduce the disincentive for transplant centers and increase access to life-saving kidney transplantation in the SCD patient population [45,46].

Our study has some limitations. Firstly, information on non-renal SCD-related organ injuries/complications was not available to review in the OPTN/UNOS database, thus we were unable to account for the effects of severity of SCD on transplant outcomes. Secondly, data on SCD genotype, comorbidity of SCD recipients, the frequency of vaso-occlusive crises, blood transfusion, the development of de novo HLA-DSA and RBC alloantibodies, treatment regimen for SCD, causes of death or kidney allograft failure in SCD recipients were unavailable. Consequently, despite adjusting for available confounders, we determined that patient survival was not significantly different between the recent era (2010–2019) and the early era (2000–2009), but we did not have information to understand the reason for the lack of improvement in post-transplant outcomes in SCD recipients. With the above noted limitations on SCD severity data, we cannot conclude that there was no difference in the advances of SCD kidney transplant recipients’ care. Physicians may have a lower threshold to offer kidney transplantation to SCD patients with more SCD-related organ involvements in the recent era, which could have affected patient outcomes. Future studies evaluating the effects of SCD severity and frequency of vaso-occlusive crises on outcomes of kidney transplant recipients with SCD are required. Finally, there are limited data on emerging treatments for SCD in the OPTN/UNOS database. Bone marrow transplantation has recently emerged as a novel treatment for SCD [49], and the beneficial effects of bone marrow transplantation in improving target organ damage have been reported [12,50,51]. Nevertheless, successful bone marrow transplant has only been reported in a limited number of dialysis and kidney transplant patients with SCD [50,52,53]. Additional studies are needed to assess its potential effects on clinical outcomes of SCD kidney transplant recipients [12]. In addition, while genetic approaches against SCD such as CRISPR gene correction therapy are promising, investigation in this area is still in the early stages [54].

## 5. Conclusions

In conclusion, patient and allograft survival in SCD kidney recipients were worse than that of recipients with other diagnoses. Overall SCD patient and allograft outcomes in the recent era did not improve from the early era. The findings of our study should not discourage kidney transplantation for ESKD patients with SCD due to the known survival benefits of transplantation compared with remaining on dialysis. Future studies are urgently needed to identify strategies to improve patient and allograft survival in SCD kidney recipients. In addition, it is reasonable to assign risk adjustment for SCD patients.

## Figures and Tables

**Figure 1 jcm-10-03063-f001:**
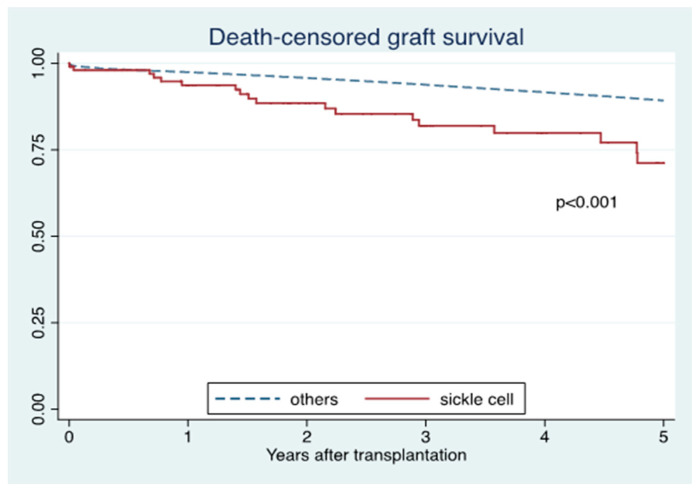
Death-censored graft survival of sickle cell disease compared to non-sickle cell disease recipients.

**Figure 2 jcm-10-03063-f002:**
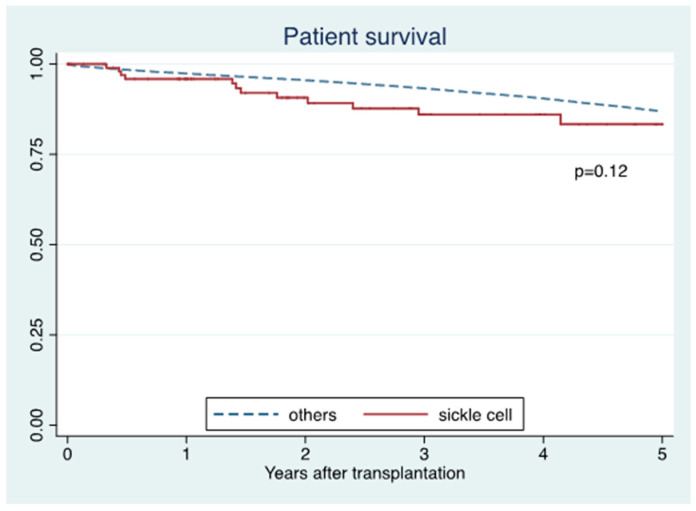
Patient survival of sickle cell disease compared to non-sickle disease recipients.

**Figure 3 jcm-10-03063-f003:**
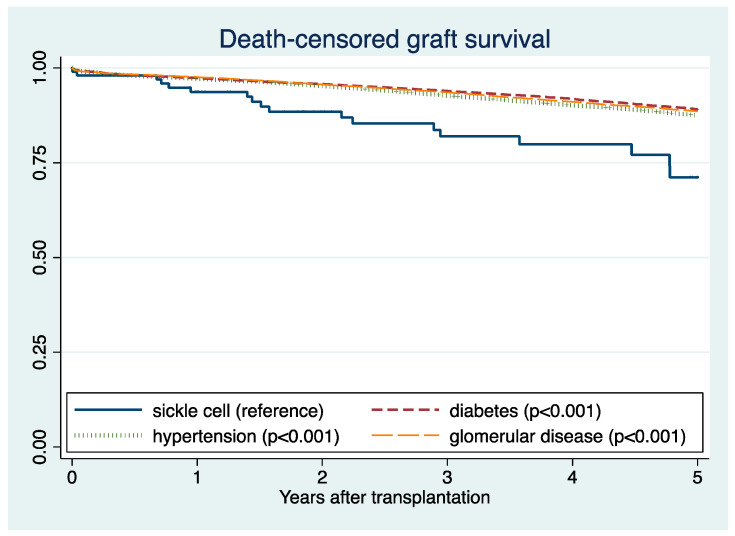
Death-censored graft survival according to the causes of end-stage kidney disease.

**Figure 4 jcm-10-03063-f004:**
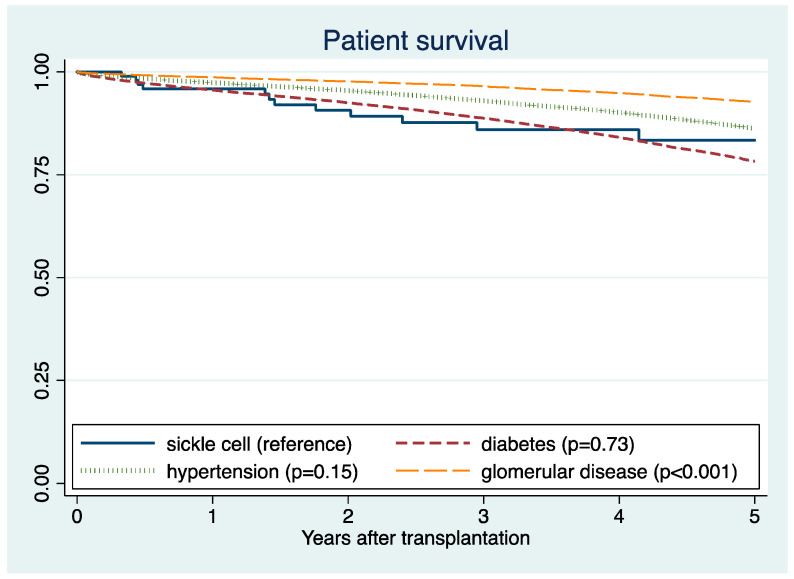
Patients survival according to the causes of end-stage kidney disease.

**Figure 5 jcm-10-03063-f005:**
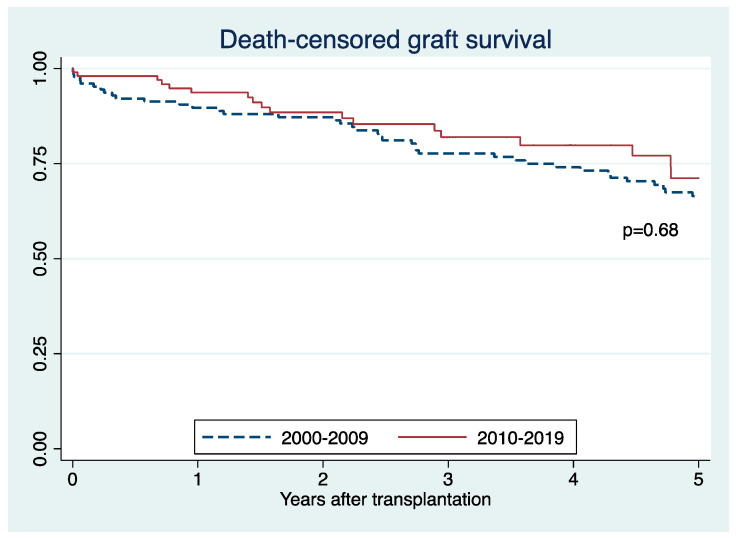
Death-censored graft survival of sickle cell disease recipients in 2010–2019 and 2000–2009.

**Figure 6 jcm-10-03063-f006:**
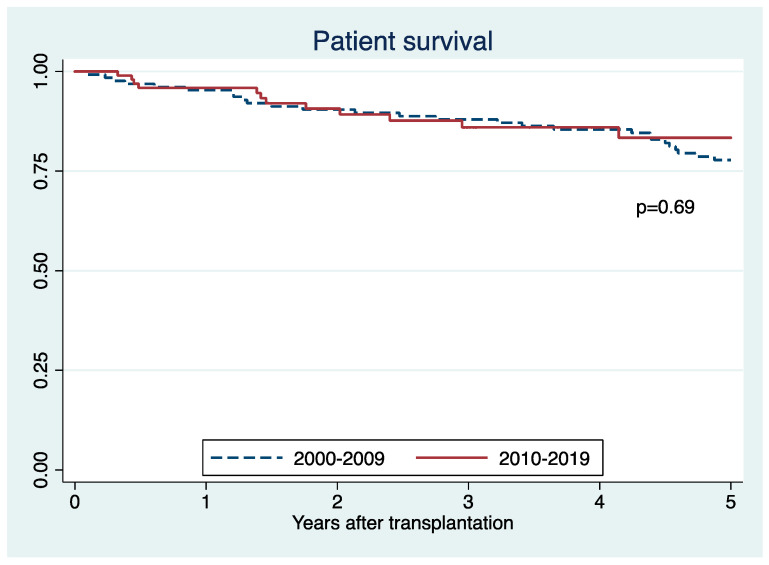
Patient survival of sickle cell disease recipients in 2010–2019 and 2000–2009.

**Table 1 jcm-10-03063-t001:** Clinical characteristics between sickle cell disease and non-sickle cell disease kidney transplant recipients.

Characteristics	Sickle Cell Disease (*n* = 105)	Non-Sickle Cell Disease (*n* = 146,325)	*p*-Value
Recipient age (year), median (25th, 75th)	41 (33, 51)	53 (41, 63)	<0.001
Male, %	47.6	61.0	0.005
African American, %	93.3	26.9	<0.001
Recipient BMI (kg/m^2^), median (25th, 75th)	22.7 (20.4, 27.1)	27.9 (24.0, 32.1)	<0.001
Living-donor kidney transplants, %	28.6	34.1	0.24
Dialysis duration (%)			
Preemptive <1 years 1–3 years >3 years Missing	7.6 13.3 20.0 57.1 1.9	19.2 14.2 23.6 42.1 0.9	0.003 0.80 0.39 0.002 0.28
Diabetes, %	1.0	33.4	<0.001
PRA (%)			
<20 20–70 >70 Missing	62.9 18.1 15.2 3.8	76.7 12.4 9.3 1.6	0.001 0.08 0.035 0.07
ABO incompatible	1.0	1.2	0.84
HLA mismatches, median (25th, 75th)	4 (3, 5)	4 (3, 5)	0.44
Donor age (year), median (25th, 75th)	30 (24, 45)	41 (28, 52)	<0.001
Donor race, %			
White Black Hispanic Others	54.3 28.6 14.3 2.9	68.4 12.6 14.2 4.8	0.002 <0.001 0.98 0.35
KDPI, median (25th, 75th)	39 (11, 60)	44 (23, 67)	0.008
Induction therapy, %			
Thymoglobulin Alemtuzumab Basiliximab Other induction No induction	66.7 8.6 24.8 3.8 8.6	52.7 15.8 23.5 1.9 9.6	0.004 0.04 0.76 0.15 0.72
Maintenance therapy, %			
Tacrolimus Cyclosporine Mycophenolate Azathioprine mTOR inhibitors Steroids	90.5 1.9 94.3 0 0 70.5	91.6 2.0 92.9 0.4 1.1 65.8	0.68 0.96 0.57 0.50 0.29 0.31

Abbreviation: BMI, body mass index; KDPI, kidney donor profile index; mTOR inhibitors, the mammalian target of rapamycin inhibitors; PRA, panel reactive antibody.

**Table 2 jcm-10-03063-t002:** Post-transplant outcomes of sickle cell disease compared to non-sickle cell disease.

(A) Death-Censored Graft Failure				
	Univariate Model		Multivariate Model *	
	HR (95% CI)	*p*-Value	HR (95% CI)	*p*-Value
Recent era (2010–2019)				
Sickle cell versus non-sickle cell	2.92 (1.92–4.44)	<0.001	1.98 (1.30–3.01)	0.001
Sickle cell versus diabetes	2.84 (1.87–4.32)	<0.001	2.32 (1.52–3.56)	<0.001
Sickle cell versus hypertension	2.53 (1.66–3.84)	<0.001	1.91 (1.25–2.92)	0.003
Sickle cell versus glomerular disease	2.80 (1.84–4.25)	<0.001	1.91 (1.25–2.92)	0.003
Sickle cell between 2000–2009 and 2010–2019	0.90 (0.54–1.50)	0.68	0.99 (0.57–1.73)	0.98
**(B) Mortality**				
	**Univariate Model**		**Multivariate Model ***	
	**HR (95% CI)**	***p*-Value**	**HR (95% CI)**	***p*-Value**
Recent era (2010–2019)				
Sickle cell versus others	1.67 (1.03–2.73)	0.039	2.87 (1.75–4.68)	<0.001
Sickle cell versus diabetes	0.95 (0.58–1.54)	0.82	1.96 (1.20–3.22)	0.007
Sickle cell versus hypertension	1.61 (0.99–2.64)	0.056	3.08 (1.88–5.04)	<0.001
Sickle cell versus glomerular disease	3.18 (1.95–5.21)	<0.001	3.41 (2.07–5.61)	<0.001
Sickle cell between 2000–2009 and 2010–2019	0.89 (0.50–1.59)	0.70	0.93 (0.50–1.74)	0.82

* Adjusted for recipient age, sex, race, BMI, dialysis status, cPRA, donor age, donor race, KDPI, and induction therapy.

**Table 3 jcm-10-03063-t003:** Clinical characteristics between sickle cell disease in 2010–2019 and 2000–2009.

Characteristics	Sickle Cell Disease in 2010–2019 (*n* = 105)	Sickle Cell Disease in 2000–2009 (*n* = 128)	*p*-Value
Recipient age (year), median (25th, 75th)	41 (33, 51)	35 (29, 45)	<0.001
male, %	47.6	58.6	0.10
African American, %	93.3	91.4	0.58
Recipient BMI (kg/m^2^), median (25th, 75th)	22.7 (20.4, 27.1)	20.5 (18.1, 23.7)	<0.001
Living-donor kidney transplants, %	28.6	34.4	0.34
Dialysis duration (%)			
Preemptive <1 years 1–3 years >3 years Missing	7.6 13.3 20.0 57.1 1.9	7.8 18.0 25.8 38.3 10.2	0.96 0.34 0.30 0.004 0.01
Diabetes, %	1.0	1.6	0.68
PRA (%)			
<20 20–70 >70 Missing	62.9 18.1 15.2 3.8	57.8 15.6 11.7 14.8	0.43 0.62 0.43 0.005
ABO incompatible	1.0	0	0.27
HLA mismatches, median (25th, 75th)	4 (3, 5)	4 (3, 5)	0.33
Donor age (year), median (25th, 75th)	30 (24, 45)	39 (27, 47)	0.03
Donor race, %			
White Black Hispanic Others	54.3 28.6 14.3 2.9	46.9 37.5 12.5 3.1	0.26 0.15 0.69 0.90
KDPI, median (25th, 75th)	39 (11, 60)	46 (30, 66)	0.02
Induction therapy, %			
Thymoglobulin Alemtuzumab Basiliximab Other induction No induction	66.7 8.6 24.8 3.8 8.6	37.5 5.5 19.5 10.2 29.7	<0.001 0.35 0.34 0.06 <0.001
Maintenance therapy, %			
Tacrolimus Cyclosporine Mycophenolate Azathioprine mTOR inhibitors Steroids	90.5 1.9 94.3 0 0 70.5	71.1 22.7 81.3 3.9 14.1 82.0	<0.001 <0.001 0.003 0.04 <0.001 0.04

## Data Availability

Data is available upon reasonable request to corresponding author.

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
