# Peer review of "Outcomes of Kidney Transplant Recipients with Sickle Cell Disease: An Analysis of the 2000–2019 UNOS/OPTN Database"

_jcm, 2021, doi:10.3390/jcm10143063_

Round 1

Reviewer 1 Report

Summary: This article explores the underreported data of kidney transplant in patients with sickle cell disease. The authors take on the task of comparing recent renal transplant data in SCD patients to transplant patients without SCD, demonstrating that graft survival and mortality are lower in SCD patients than their non-SCD transplanted counterparts. Further, the authors show that there is no difference in graft or overall survival for SCD patients after renal transplant in the recent period compared to procedures performed >10year prior. The study seems to recapitulate early work comparing SCD to non-SCD transplant outcomes and concludes that similar trends continue: SCD patients fare worse than non-SCD patients and there is no difference between current and ‘early’ transplants for SCD patients.

Comments/Suggestions:

-This work very clearly presents the existing data but fails to convey to the audience its significance. The introduction alludes to the past 10-years of transplant advances. The authors seem to hypothesize that because transplant care and outcomes are better there will be a difference in outcomes in the SCD population but do not discuss any of the general transplant improvements or specific discoveries that may be relevant to the SCD population. A more developed introduction and hypothesis which discusses more specifically what is different in the ‘recent era’ for all transplants as well as for SCD patients (i.e. degree of HLA/epitope matching, infection prevention, emerging immunosuppressing agents, standardization of pre-transplant preparation for SCD patients, use of post-transplant SCD disease modifying therapy, pre-transplant APOL1 screening) would help contextualize the primary question. The authors should specifically discuss recommendations for pre and peri-transplant care for SCD patients including HU use and use of exchange transfusions.

-additional references to include:

+the impact of exchange transfusion for patients with SCD after kidney transplant https://pubmed.ncbi.nlm.nih.gov/32790687/

+"how I treat renal complications in SCD" https://pubmed.ncbi.nlm.nih.gov/24764565/

-please provide more information about the study population, to include ages eligible and SCD genotypes included. Did any SCD patients have co-morbid HTN?

-is there any descriptive information available about causes of death in the SCD population? Particularly given that the average life expectancy for adults with SCD is ~42, which is close to the median age at time of kidney transplant in the SCD group

-previous data (Ojo, 1999) suggests that there may be a short term survival benefit for transplanted SCD patients. I would suggest an additional analysis of 1-year graft and overall survival to explore short vs. long term outcomes in the modern era.

-kidney donors of African American race have demonstrably poorer outcomes; can you include the race of the kidney donor?

-the authors suggest that “ongoing unequal access to kidney transplantation for SCD is a reflection of the transplant community’s concerns related to increased mortality risk for SCD patients.” The authors would do well to at least briefly acknowledge in the discussion the historical role of racial health disparities and comment specifically that worse outcomes in SCD patients persisted even when controlling for race in the multivariate analysis. Wang et al published a nice historical review of this ‘blind spot’ which may be worth noting https://pubmed.ncbi.nlm.nih.gov/30777381/

Author Response

Response to Reviewer#1

This article explores the underreported data of kidney transplant in patients with sickle cell disease. The authors take on the task of comparing recent renal transplant data in SCD patients to transplant patients without SCD, demonstrating that graft survival and mortality are lower in SCD patients than their non-SCD transplanted counterparts. Further, the authors show that there is no difference in graft or overall survival for SCD patients after renal transplant in the recent period compared to procedures performed >10year prior. The study seems to recapitulate early work comparing SCD to non-SCD transplant outcomes and concludes that similar trends continue: SCD patients fare worse than non-SCD patients and there is no difference between current and ‘early’ transplants for SCD patients.

Response: We thank you for reviewing our manuscript and for your critical evaluation.

Comment #1

This work very clearly presents the existing data but fails to convey to the audience its significance. The introduction alludes to the past 10-years of transplant advances. The authors seem to hypothesize that because transplant care and outcomes are better there will be a difference in outcomes in the SCD population but do not discuss any of the general transplant improvements or specific discoveries that may be relevant to the SCD population. A more developed introduction and hypothesis which discusses more specifically what is different in the ‘recent era’ for all transplants as well as for SCD patients (i.e. degree of HLA/epitope matching, infection prevention, emerging immunosuppressing agents, standardization of pre-transplant preparation for SCD patients, use of post-transplant SCD disease modifying therapy, pre-transplant APOL1 screening) would help contextualize the primary question. The authors should specifically discuss recommendations for pre and peri-transplant care for SCD patients including HU use and use of exchange transfusions.

-additional references to include:

+the impact of exchange transfusion for patients with SCD after kidney transplant https://pubmed.ncbi.nlm.nih.gov/32790687/

+"how I treat renal complications in SCD" https://pubmed.ncbi.nlm.nih.gov/24764565/

Response: We agree with the reviewer and appreciate these important points to help improve our manuscript. We have these important points as reviewer’s suggestions including differences in prior era and recent era for all transplants as well as for SCD patients. In addition, we also additionally discussed on pre and peri-transplant care for SCD patients including HU use and use of exchange transfusions, as reviewer’s suggestion. We also found the suggested references very helpful and used as our updated reference#17 (PMID: 32790687) and reference#18 (PMID: 24764565).

The following text has been added in the introduction and discussion.

“Immunosuppression and kidney transplant care have significantly advanced for the past decades [14], and overall kidney transplantation outcomes have significant improved in the recent era [15,16]. In addition to emerging immunosuppressing agents, advances in human leukocyte antigen (HLA)/epitope matching, infection prevention, pre-transplant apolipoprotein L1 gene (APOL1), screening standardization of pre-transplant preparation for SCD patients, use of post-transplant SCD disease modifying therapy may impact transplant outcomes of patients with SCD [17-19]

“Although data on the effects of hydroxyurea on kidney transplant allograft are limited, the use of hydroxyurea has shown potential benefits in reduction of proteinuria, stabilization in kidney function, and reduced mortality risk among non-kidney transplant SCD patients [18,28-30]. While early posttransplant blood transfusion may increase immunological risk and increased risk of rejection, especially in under-immunosuppressed recipients [31], recent study demonstrated that blood transfusion while on adequate immunosuppressive regimens within 1-week post-kidney transplantation is safe without significant association with de novo HLA-DSA development [32]. Among kidney transplant recipients with SCD, a well-conducted multicenter study on 34 SCD kidney transplant recipients between 1997 and 2017 across 6 London hospitals ensured the safety of long-term automated exchange blood transfusions (EBT) with improvement in allograft and patient outcomes and no increase in antibody formation or allograft rejection [17]. However, data on the use of hydroxyurea and blood transfusion are limited in the OPTN/UNOS database. Thus, future studies are needed to assess the impact of hydroxyurea, EBT, and perioperative management on outcomes of kidney transplant recipients with SCD.”

Comment #2

please provide more information about the study population, to include ages eligible and SCD genotypes included. Did any SCD patients have co-morbid HTN?

Response:  The reviewer raises important point. We did not limit age in eligibility criteria, and therefore, included both pediatric and adult patients in this study. The following statements have been revised to clarify age eligibility

“The primary cohort includes pediatric and adult solitary first kidney transplant recipients occurring from 2010 to 2019.”

We did not have data on SCD genotype or comorbidity of SCD recipients in the UNOS database. The followings statements have been added in the limitation.

“Secondly, data on SCD genotype, comorbidity of SCD recipients, the frequency of vaso-occlusive crises, blood transfusion, the development of de novo HLA-DSA and RBC alloantibodies, treatment regimen for SCD, cause of death or kidney allograft failure in SCD recipients was unavailable. Consequently, despite adjusting for available confounders, we determined that patient survival was not significantly different between the recent era (2010-2019) and the early era (2000-2009) but we did not have information to understand the reason for the lack of improvement in post-transplant outcomes in SCD recipients.”

Comment #3

is there any descriptive information available about causes of death in the SCD population? Particularly given that the average life expectancy for adults with SCD is ~42, which is close to the median age at time of kidney transplant in the SCD group

Response: We appreciate the reviewer’s important point. OPTN/UNOS did not have the data on causes of death and kidney allograft failure in SCD recipients. The following statements have been added to address this limitation.

“Secondly, data on SCD genotype, comorbidity of SCD recipients, the frequency of vaso-occlusive crises, blood transfusion, the development of de novo HLA-DSA and RBC alloantibodies, treatment regimen for SCD, causes of death or kidney allograft failure in SCD recipients was unavailable. Consequently, despite adjusting for available confounders, we determined that patient survival was not significantly different between the recent era (2010-2019) and the early era (2000-2009) but we did not have information to understand the reason for the lack of improvement in post-transplant outcomes in SCD recipients.”

Comment #4

previous data (Ojo, 1999) suggests that there may be a short-term survival benefit for transplanted SCD patients. I would suggest an additional analysis of 1-year graft and overall survival to explore short vs. long term outcomes in the modern era.

Response: We agree with this important point. We thus performed additional analysis as suggested. The following statements have been added for additional analysis of 1-year graft and patient survival.

“One-year death-censored graft survival in SCD was lower than non-SCD recipients (94% vs. 98%; p=0.02), whereas there was no significant difference in patient survival between SCD and non-SCD recipients (96% vs. 97%; p=0.36). In adjusted analysis, SCD recipients was significantly associated with higher 1-year death-censored graft failure (HR 2.22; 95% CI 1.01-4.95). SCD recipients had higher 1-year death than non-SCD recipients with HR of 2.24 although it did not reach statistical significance (p=0.11) due to small number of deaths within 1 years after kidney transplant.”

Comment #5

kidney donors of African American race have demonstrably poorer outcomes; can you include the race of the kidney donor?

Response: We agree with this important point. The data on donor race has been added in revised Table 1 and Table 3, as suggested. We also included donor race in adjusted analysis.

Comment #6

the authors suggest that “ongoing unequal access to kidney transplantation for SCD is a reflection of the transplant community’s concerns related to increased mortality risk for SCD patients.” The authors would do well to at least briefly acknowledge in the discussion the historical role of racial health disparities and comment specifically that worse outcomes in SCD patients persisted even when controlling for race in the multivariate analysis. Wang et al published a nice historical review of this ‘blind spot’ which may be worth noting https://pubmed.ncbi.nlm.nih.gov/30777381/

Response: We agree with this important point. We have included this important point and found the suggested reference by Wang et al helpful and used as our updated reference#44 (PMID: 30777381)

“This is potentially due to increased risk for graft loss and mortality after transplantation [8,10], or racial disparities in kidney transplantation [42,43], representing a potential "blind spot" regarding racial disparities in access and health outcomes [44].”

Reviewer 2 Report

Intro:

Line 55: “Almost half of SCD patients develop chronic kidney disease; the prevalence of end stage kidney disease (ESKD) ranges from 5% up to 18% with a median age of 23 years [1,5-7].”

--true for those with hbSS but not “all types” of SCD.  Needs modification

Line 56: Despite its prevalence, the overall number of people with SCD who go on to receive a kidney transplant remain low [8-10]

--also should be rewritten as “despite the prevalence of ESKD in SCD, few of these individuals” “receive” or “are approved for” as opposed to “go on”

Line 74: “We conducted this study using the UNOS database with the aim to 1) 74 assess patient and allograft survival between recipients with SCD vs. non-SCD in the recent era (2010-2019), and to 2) examine outcomes of SCD recipients in the recent era (2010-2019), compared to the early era (2000-2009).”

--should say "compare" not assess

Data Collection:

Line 92: “The following variables were extracted: recipient age, sex, race, body mass index (BMI), transplant type, dialysis duration, history of diabetes mellitus, panel reactive antibody (PRA) level, donor age, kidney donor profile index (KDPI), induction and maintenance therapy. Missing data was not imputed.

--Assuming the authors also collected sickle cell disease status?  Not listed.

--Once listed, how was diagnosis of SCD confirmed in the database? 

Other questions:

  1. Was data available on RBC or HLA antibodies? This should be included or information about why this is not included should also be written.
  2. Similarly, the concern for RBC alloimmunization and its effect on graft survival is also not included in the discussion.

Author Response

Comment #1

Line 55: “Almost half of SCD patients develop chronic kidney disease; the prevalence of end stage kidney disease (ESKD) ranges from 5% up to 18% with a median age of 23 years [1,5-7].”

--true for those with hbSS but not “all types” of SCD.  Needs modification

Response: We appreciate the reviewer’s expertise. We agree with the reviewer and thus, have made correction on the sentence as suggestion.

“Almost half of SCD patients with HbSS genotype develop chronic kidney disease; the prevalence of end stage kidney disease (ESKD) ranges from 5% up to 18% with a median age of 23 years [1,5-7].”

Comment #2

Line 56: Despite its prevalence, the overall number of people with SCD who go on to receive a kidney transplant remain low [8-10]

--also should be rewritten as “despite the prevalence of ESKD in SCD, few of these individuals” “receive” or “are approved for” as opposed to “go on”

Response: We thank you for reviewing our manuscript and for your critical evaluation. We appreciate the reviewer’s comment. We agree and thus have revised this statement as the reviewer’s suggestion.

"Despite the prevalence of ESKD in SCD, few of these individuals receive a kidney transplant [8-10]”

Comment #3

Line 74: “We conducted this study using the UNOS database with the aim to 1) 74 assess   patient and allograft survival between recipients with SCD vs. non-SCD in the recent era (2010-2019), and to 2) examine outcomes of SCD recipients in the recent era (2010-2019), compared to the early era (2000-2009).”

Response: We appreciate the reviewer’s thorough review. We agree with the reviewer. We have changed “assess” to “compare” as suggested.

“We conducted this study using the UNOS database with the aim to 1) compare patient and allograft survival between recipients with SCD vs. non-SCD in the recent era (2010-2019), and to 2) examine outcomes of SCD recipients in the recent era (2010-2019), compared to the early era (2000-2009).”

Comment #4

Line 92: “The following variables were extracted: recipient age, sex, race, body mass index (BMI), transplant type, dialysis duration, history of diabetes mellitus, panel reactive antibody (PRA) level, donor age, kidney donor profile index (KDPI), induction and maintenance therapy. Missing data was not imputed.

--Assuming the authors also collected sickle cell disease status?  Not listed.

--Once listed, how was diagnosis of SCD confirmed in the database? 

Response: The reviewer raises important point. The following statements have been revised to add sickle status in the list of variables we abstracted from the database as well as to clarify the diagnosis of sickle cell disease.

“The following variables were extracted: recipient age, sex, race, body mass index (BMI), transplant type, dialysis duration, causes of end-stage kidney disease, SCD status, history of diabetes mellitus, panel reactive antibody (PRA) level, donor age, kidney donor profile index (KDPI), induction and maintenance therapy. SCD as the cause of ESRD was reported by individual transplant center to OPTN/UNOS database.”

Comment #5

Was data available on RBC or HLA antibodies?  This should be included or information about why this is not included should also be written.

Response: We appreciate the reviewer’s important comment. Data on RBC or HLA antibodies are not available in the UNOS database. However, this is very good point, and thus we additionally included available data from UNOS database including the data on ABO incompatibility and HLA mismatch has been added in Table 1 and  Table 3, as suggested.

Comment #6

Similarly, the concern for RBC alloimmunization and its effect on graft survival is also not included in the discussion.

Response: We appreciate the reviewer’s important comment. We agree and thus additionally discussed on the point of blood transfusion, RBC alloimmunization and allograft survival among SCD recipients as reviewer’s suggestions.

“While early posttransplant blood transfusion may increase immunological risk and increased risk of rejection, especially in under-immunosuppressed recipients [31], recent study demonstrated that blood transfusion while on adequate immunosuppressive regimens within 1-week post-kidney transplantation is safe without significant association with de novo HLA-DSA development [32]. Among kidney transplant recipients with SCD, a well-conducted multicenter study on 34 SCD kidney transplant recipients between 1997 and 2017 across 6 London hospitals ensured the safety of long-term automated exchange blood transfusions (EBT) with improvement in allograft and patient outcomes and no increase in antibody formation or allograft rejection [17]. However, data on the use of hydroxyurea and blood transfusion are limited in the OPTN/UNOS database. Thus, future studies are needed to assess the impact of hydroxyurea, EBT, and perioperative management on outcomes of kidney transplant recipients with SCD.”

We greatly appreciated the editor and reviewer’s time and comments to improve our manuscript.

Round 2

Reviewer 1 Report

Only one minor suggestion: the authors should define the abbreviation DSA (page 11, line 225). Otherwise the comments have been adequately addressed.

Author Response

Response to Reviewer#1

Only one minor suggestion: the authors should define the abbreviation DSA (page 11, line 225). Otherwise the comments have been adequately addressed.

Response: We thank you for reviewing our manuscript and for your critical evaluation. We agree with the reviewer and have now defined the abbreviation of DSA (page 11, line 225) as suggested.

We greatly appreciated the editor and reviewer’s time and comments to improve our manuscript.

Reviewer 2 Report

Few comments only.  Great improvements.  

Main issues:

  1. The results (especially in the first paragraph) go back and forth in describing death censored graft survival and death censored graft failure which is confusing and can be improved
  2. Discussion needs to be a little more streamlined.  You are saying that a. Patients with DM do better b/c of better DM care b. Patients with SCD still have SCD which may or may not impact survival (if the survival is better in the London paper say it here) and c. More people with SCD likely deserve to get to transplant in a multi-center system.  Make sure your comments line up in this was (currently you start with SCD and severity, go to DM, go back to SCD problems etc). 

Author Response

Response to Reviewer#2

 Few comments only.  Great improvements. 

Response: We thank you for reviewing our manuscript and for your critical evaluation.

Comment 1.The results (especially in the first paragraph) go back and forth in describing death censored graft survival and death censored graft failure which is confusing and can be improved

Response: We agree with the reviewer and have now revised our result section as suggested. The following text has been revised as reviewer’ suggestion.

“One-year death-censored graft survival in SCD was lower than non-SCD recipients (94% vs. 98%; p=0.02), whereas there was no significant difference in patient survival between SCD and non-SCD recipients (96% vs. 97%; p=0.36). In adjusted analysis, SCD recipients was significantly associated with lower 1-year death-censored graft survival (HR 2.22; 95% CI 1.01-4.95). SCD recipients had lower 1-year patient survival than non-SCD recipients with HR of 2.24 although it did not reach statistical significance (p=0.11) due to small number of deaths within 1 years after kidney transplant.

Five-year death-censored graft survival in SCD was lower than non-SCD recipients (71% vs. 89%; p<0.001) (Figure 1), whereas five-year patient survival was comparable between two groups (83% vs. 87%; p=0.12) (Figure 2). In adjusted analysis, SCD recipients was significantly associated with lower death-censored graft survival (HR 1.98; 95% CI 1.30-3.01, p=0.001) (Table 2A) and patient survival (HR 2.87; 95% CI 1.75-4.68, p<0.001), compared to non-SCD recipients (Table 2B).”

Comment 2. Discussion needs to be a little more streamlined.  You are saying that a. Patients with DM do better b/c of better DM care b. Patients with SCD still have SCD which may or may not impact survival (if the survival is better in the London paper say it here) and c. More people with SCD likely deserve to get to transplant in a multi-center system.  Make sure your comments line up in this was (currently you start with SCD and severity, go to DM, go back to SCD problems etc).

Response: We appreciate the reviewer’s input to improve our revised manuscript. We agree with the reviewer and have now revised our discussion section as suggested. The following text has been revised as reviewer’ suggestion.

“Given there is no difference between patient survival among SCD in years 2010-2019 when compared to years 2000-2009, this finding likely reflects a significant improvement in diabetes care post kidney transplantation for the past decade [42-44]. Kidney transplant recipients with SCD still have complications of SCD post-transplant [20-22].  Since SCD is a systemic disease, it can result in extra-renal manifestations, such as pulmonary hypertension, thrombotic events, infections, cardiomyopathy and cirrhosis [40,41]. Unfortunately, there seems to bethe findings from our study suggest a lack in similar improvement in SCD care post transplantation when compared to other causes of ESKD including diabetes. Future studies are required to identify effective strategies to improve care for kidney transplant recipients with SCD such as care in a multi-center system and long-term follow-up in tertiary care centers.” 

We greatly appreciated the editor and reviewer’s time and comments to improve our manuscript.
